# Correlation between Core Stability and Upper-Extremity Performance in Male Collegiate Athletes

**DOI:** 10.3390/medicina58080982

**Published:** 2022-07-23

**Authors:** Shibili Nuhmani

**Affiliations:** Department of Physical Therapy, College of Applied Medical Sciences, Imam Abdulrahman Bin Faisal University, P.O. Box 2435, Dammam 31451, Saudi Arabia; snuhmani@iau.edu.sa; Tel.: +966-554270531

**Keywords:** core muscles, upper limb performance, McGill test, double-leg lowering

## Abstract

*Background:* The purpose of this study was to investigate the correlation of core stability, as measured by the McGill and double-leg lowering (DLL) test, with upper-extremity performance, as measured by the upper-quarter Y-balance test (UQYBT), medicine ball throw test (MBTT) and functional throwing performance index (FTPI) test, in collegiate athletes. *Materials and Methods*: A sample of 61 collegiate athletes from Imam Abdulrahman Bin Faisal University participated in the study. Their core stability was assessed through their McGill and DLL test scores. Their upper-extremity performance was assessed through their UQYBT, MBTT and FTPI test scores. *Results*: The McGill test score had a significant strong positive correlation with the MBTT score (*p* = 0.02, r = 0.67) and a significant moderate positive correlation with the UQYBT score (*p* = 0.01, r = 0.46). There was no significant correlation between the McGill and FTPI test scores (*p* ≥ 0.05). The DLL test score was positively correlated with the MBTT score (*p* = 0.02, r = 0.25) but had no significant correlation with the other sports performance variables (*p* ≥ 0.05). *Conclusion*: The study results suggest that core stability measures are positively correlated with most of the upper-extremity athletic-performance measures in collegiate athletes. The MBTT score was found to be the most significantly correlated with the scores in both core stability tests among all the upper-extremity athletic-performance tests in this study. However, due to the nature of this study, a cause–effect relationship cannot be established on the basis of the study’s findings, and the study results should be interpreted with caution.

## 1. Introduction

Core stability is the capability of controlling the position and motion of the trunk with the pelvis for optimum force generation, transfer and absorption from the proximal and distal segments during various activities [1,2]. The core is crucial in distributing forces between the torso and the extremities. Higher levels of core stability allow athletes to perform quick, coordinated movements, which enhances their athletic performance. A positive relationship between core stability and athletic performance is being documented in sports such as soccer [3,4,5], hockey [6], tennis [7], badminton [8], basketball [9] and lacrosse [10]. Core training has, thus, become a key component of any athletic training program for enhancing the performance of athletes and preventing injuries.

While most of the previous studies that focused on the relationship between core stability and athletic performance focused on lower-extremity performance, a few studies with contradictory results explored the core’s influence on upper-extremity performance. Some of these studies indicated that core strength and core endurance are not strong predictors of the upper extremities’ athletic performance [11]. Chaudhari, et al. [12] reported a strong positive correlation between core stability and pitching performance in professional baseball pitchers. A recent study reported a fair, positive correlation between core stability and medicine ball throw test (MBTT) performance in badminton players [8]. Another study reported a weak to moderate correlation between core stability measures and the backward overhead MBTT score [13]. At the same time, Sharrock, et al. [14] reported a negative correlation between the MBTT and core stability test scores in student athletes. Söğüt [15] did not find any correlation between core stability and tennis-related performance parameters in competitive junior tennis players. According to Silfies, Ebaugh, Pontillo and Butowicz [11], a true causal relationship between core training and upper-extremity performance cannot be inferred from any of the previous studies.

It has been suggested that studies in this field should first focus on determining whether core stability measures are correlated with upper-extremity athletic-performance indicators. Therefore, this study aimed to investigate the correlation between core stability, as measured by the McGill and double-leg lowering (DLL) test scores, and upper-extremity performance, as measured by the upper-quarter Y-balance test (UQYBT), MBTT and functional throwing performance index (FTPI) test scores, in collegiate athletes. The study’s hypothesis is that there is a correlation exist between core stability, as measured by the McGill and DLL test scores, and upper-extremity performance, as measured by the UQYBT, MBTT and FTPI test scores, in collegiate athletes. Finding these correlations will greatly impact sports rehabilitation and sports performance research and will serve as a stepping stone for sports-specific exercise prescriptions and for identifying the appropriate training protocol for athletes.

## 2. Materials and Methods

### 2.1. Sample

The volunteer participants in the current study were 61 healthy male collegiate athletes between the ages of 18and 30 years from Imam Abdulrahman Bin Faisal University (IAU) in Dammam. The descriptive characteristics, such as age, height, body mass, body mass index and weekly training hours, were recorded. The testing was performed at the physical-therapy laboratory of IAU from 8 to 11 a.m. The sample size was calculated as 61 using a sample size calculator (https://www.ai-therapy.com/psychology-statistics/sample-size-calculator, accessed on 15 December 2021) and on the basis of a previous study on the correlation of core stability with athletic performance in collegiate athletes, with a 0.268 correlation coefficient, a 0.05 significance level and 0.8 statistical power. Those who (1) had experienced any abdominal or musculoskeletal injuries/pain that required medical attention for the past 2 months, (2) had any systemic or metabolic disease, neurological disorder or biomechanical or postural abnormality and (3) were taking any medication that could affect the testing were excluded from the current study. The researcher explained the purpose and protocol of the study to all the participants prior to the study’s commencement, and the participants’ written informed consent was then obtained. The study was approved by the institutional ethical committee of IAU (Approval number- IRB-2022-03-107) and was carried out in compliance with the Declaration of Helsinki.

### 2.2. Procedure

The design of the study was a multivariate correlational design. The variables for core stability included McGill and DLL test scores and the variables for upper-extremity performance included the test scores of UQYBT, MBTT and FTPI.

The tests were conducted at five stations in two different sessions, with a 24-h gap between the sessions. The tests for core stability were performed in the first session, and the tests for upper limb performance were taken in the second session. All the tests were performed in a random order in each session. All the participants completed both sessions without experiencing any pain or injury. Before the testing, the participants were asked to do a 10-min warmup consisting of light jogging and whole-body stretching. The examiners at each station first described the testing procedure to the participants with the help of a video clip, and a practice trial was performed afterwards to familiarise the participants with the tests. During the practice trial, the participants were told not to make any maximal exertion. The examiners were instructed not to give any encouragement or feedback to the participants about their performance, apart from instructions to correct the testing technique in cases in which the participants were carrying it out improperly. A 4-min rest was given between the practice trial and the testing and between tests for sufficient recovery. All the tests were performed thrice and the average was taken for analysis. All the participants performed the tests cited below.

#### 2.2.1. McGill Test

This test, which was used to assess the core endurance of the athletes who participated in the current study, consisted of the following four subtests: (a) trunk flexor test; (b) trunk extensor test; (c) right flexor test and (d) left flexor test [16]. All the participants were asked to undergo a practice trial prior to the actual test for familiarisation with the test, and the maximum time that the subjects could hold their positions was recorded with a stopwatch.

The trunk flexor test began with the subject sitting with his back against a wooden plank angled at 60°, his knee flexed 90° and his arms crossing his chest. The recording of the time begun when the wooden plank was moved backwards 10 cm and was stopped when the trunk deviated from a 60° position [2,16].

Each participant performed the right and left flexion tests in a side bridge position, with one leg placed above the other, the elbow placed on the mat and the opposite arm crossing the chest, with the hand placed on the opposite shoulder. The subject was told to elevate his pelvis, making a straight line with his body. The timer was begun when the subject was in a straight-line position and was stopped when he could no longer hold this position and when his hip had lowered onto the mat [2].

For the trunk extension test, each participant was asked to lie in a prone position with the iliac crest at the edge of the table, and the upper body was cantilevered. The lower body was secured with two straps below and above the knees. The subject was then instructed to hold his body horizontally, with his arms folded across his chest, for as long as he could. The test was terminated once the subject could no longer maintain this position, and the time was recorded [2].

#### 2.2.2. Double-Leg Lowering (DLL) Test

Each participant was asked to lie supine on the testing table, with the hip joint coinciding with the goniometer grids printed on a poster board affixed to the wall beside the table. All the participants were made to undergo practice trials on how to perform the abdominal draw-in manoeuvre. A stabiliser pressure biofeedback unit inflated with 40 mm Hg was positioned below the lumbar spine. The subject’s hip was flexed 90° with knee extension, and the subject was instructed to perform the abdominal draw-in manoeuvre to begin the test. The subject was then instructed to lower his lower limbs, while maintaining cuff pressure. The degree of hip flexion where the pressure dropped by 10 mmHg was recorded using a goniometer [17].

#### 2.2.3. Medicine Ball Throw Test (MBTT)

In this test, the subject was instructed to assume a tall kneeling position (90° knee flexion with a neutral trunk) over a mat, with a medicine ball held at chest level. A 6.6 lb medicine ball was used for the test, as recommended by Stockbrugger and Haennel [18]. The subject was asked to throw the ball horizontally as far as he could using a two-handed chest pass technique [2,14]. The test was repeated three times, and the best distance covered was used for analysis.

#### 2.2.4. Upper-Quarter Y-Balance Test (UQYBT)

This test was performed using a Y-balance tool kit (FunctionalMovement.com, Danville, VA, USA), consisting of a stance platform and three pipes equipped with movable reach indicators. The pipes represented the anterior, posteromedial and posterolateral reach directions and were marked at 1 cm increments for measurement purposes. To perform UQYBT, each participant was instructed to assume a closed-chain push-up position, and his ability to reach out in three directions by pushing the reach indicators with one hand, while keeping the other hand on the stance platform and maintaining the push-up position, was assessed. The testing order started with the right hand on the stance platform, with the left hand allowed to maximally reach the medial direction (right medial reach) towards the right inferolateral and right superolateral directions. The test was repeated with the other hand. The reach distance was calculated by reading the value on the part of the tape measure that was at the position of the reach indicator. The greatest successful reach direction measure was noted (maximal reach distance). The maximal reach distance was divided by the length of the upper limbs of the subject to normalise the data [19]. The composite reach distance scores were calculated and used for analysis by taking the average of the three normalised reach distances of both limbs.

#### 2.2.5. Functional Throwing Performance Index (FPTI) Test

The participants were asked to perform the FPTI test to find out their throwing accuracy. A 30.48 × 30.48 square board was placed on the wall at 1.22 m from the floor. Each participant was requested to throw the ball (50.8 cm circumference) as fast and as precisely as he could towards the target as many times as he could within 30 s, from 4.5 m from the wall. The FTPI score was determined by dividing the total number of successful throws (throws hitting the target) by the total number of throws made [20]. The FTPI test was shown to be a reliable tool for measuring throwing performance [21].

### 2.3. Statistical Analysis

SPSS Statistics version 24.0 for Windows (IBM Corp., Armonk, NY, USA) was used to analyse the obtained data. The Shapiro–Wilk test was used to validate the data normality (*p* > 0.05). The strength of the associations between the variables was measured using the Pearson correlation coefficient. The strength of an association was interpreted as 0–0.19 (extremely weak), 0.2–0.39 (weak), 0.40–0.59 (moderate), 0.6–0.79 (strong) or 0.8–1 (extremely strong), following the guidelines described by Campbell and Swinscow [22]. The significance level was set at ≤0.05.

## 3. Results

The descriptive characteristics of the participants (age, height, body mass, body mass index and weekly training hours) are shown in Table 1.

The descriptive values of the core stability (McGill test and DLL) and upper-extremity performance test (MBTT, UQYBT and FPTI) scores are shown in Table 2.

Table 3 shows the correlation between the core stability and upper-extremity performance test scores. The McGill test score had a significant strong positive correlation with the MBTT score (*p* = 0.023, r = 0.671) and a significant moderate positive correlation with the UQYBT score (*p* = 0.01, r = 0.462). There was no significant correlation between the McGill and FTPI test scores (*p* ≥ 0.05). The DLL test score was found to be positively correlated with the MBTT score (*p* = 0.02, r = 0.25) and not to be significantly correlated with the other sports performance variables (UQYBT and FTPI) (*p* ≥ 0.05)

## 4. Discussion

The current study was conducted to determine whether there is a correlation between core stability and upper-extremity performance measures in collegiate athletes. The study showed that only the scores in a few upper-extremity performance tests were significantly correlated with core stability measures (McGill test score vs. MBTT score; McGill test score vs. UQYBT score and DLL test score vs. MBTT score).

The core acts as a stable biomechanical platform for the peripheral muscles to work. It is considered the segmental link of the kinetic chain between the upper and lower extremities and plays a key role in maintaining stability and generating forces during various upper- and lower-extremity activities [7]. In a previous study, the core muscles were reported to be activated in a feed-forward manner during upper-extremity movements in participants with high core strength [23,24]. An athlete’s core stability and strength may impact his potential to activate his muscles in a more coordinated way and to generate greater force. This may explain the better upper-extremity performance of the participants in this study with higher core stability measures. Higher spinal stability provides a stable base, which allows the force generated by the upper-extremity muscles to be more efficiently transformed into work. When the core is unstable, it will absorb the force generated; thus, less force will be transformed into work. Core stabilisation is a dynamic concept that constantly changes to meet the postural adjustments or external forces taken by the body. Higher core stability offers a platform for better force production in the upper and lower extremities [2,7]. Because the core plays such a substantial role during various activities, it makes sense to ensure its strength and stability.

The correlation observed in our study is consistent with that observed in some previous studies [13,14,25,26]. A strong correlation between core stability measures and the score in MBTT, the test for upper-extremity power, was reported in a study conducted among male and female collegiate athletes [14]. Okada, Huxel and Nesser [13] also discovered a moderate correlation between core stability and bench press performance in athletes. The results of our study also support those of the study by Lust [26], which demonstrated an improvement in FPTI and closed-kinematic-chain upper-extremity stability in baseball players following a 6-week core strengthening program. In a previous study, a combined strength training program resulted in a 4.9% increase in service velocity in nationally ranked male junior tennis players [25]. Nesser and Lee [27] also reported a moderate positive correlation of the McGill test score with bench press/body weight in female football players. The findings of the current study also confirm the results of the studies conducted by Hassan [28] and Fernandez-Fernandez and Ellenbecker [25], which reported an improvement in sports performance measures in badminton and tennis players, respectively.

The current study used UQYBT, MBTT and the FTPI test to represent the upper extremities’ performance. UQYBT has been reported as a valid and reliable tool for measuring the dynamic balance of the upper extremities [29]. It also challenges the upper extremities’ proprioception, strength and flexibility, hence allowing the assessment of the mobility and stability of the scapular and thoracic movements out of the base of support [29]. We chose the FTPI test to assess performance, as it measures throwing accuracy and is more associated with proprioception of the upper extremities than with other parameters [30]. In the current study, a medicine ball was thrown in the tall kneeling position by each study subject, and the subject was advised to maintain his position and not fall forward upon the completion of the throw. In this test, the core muscles stabilise the trunk during vigorous upper-extremity activity.

The total McGill test score is the sum of four core endurance tests that target the flexors, extensors and lateral flexors on both sides. The McGill test score showed a significant positive correlation with the MBTT and UQYBT scores. To perform the lateral trunk endurance test, the subject has to activate the local muscles of his core, predominantly the quadratus lumborum and abdominal wall [31]. In the flexor test of the McGill protocol, it is the primary trunk flexor, the rectus abdominus (a ‘global stabiliser’), that is activated [16]. In the back extensor test, the ones activated are the longissimus and multifidus, the major extensors of the spine considered ‘local stabilisers’ [16].

Another measure of core stability used in the current study, the DLL test score, showed a significant positive correlation with the MBTT score. Shields and Heiss [32] investigated the electromyographic activity of the abdominal muscles during DLL and knee curl ups. The researchers reported higher muscle activation of the rectus abdominis (external and internal oblique) in DLL than in the knee curl up. DLL demands a narrow base of support for the upper body and trunk and longer lever arms for the lower extremities, requiring higher trunk stabilisation. A similar result was reported by Sharrock, Cropper, Mostad, Johnson and Malone [14], who investigated the relation between the DLL test score and various other performance test scores and observed that the DLL test score is highly correlated with the MBTT score, compared to other performance test scores.

In the current study, both core stability parameters were significantly correlated with the medicine ball throw test, with a *p* value 0.02. A similar finding was reported by Shinkle, et al. [33], who reported a correlation between core performance variables and the medicine ball throw test among National Collegiate Athletic Association Division I football players. Hassan [28] also reported an improvement in stroke velocity following eight weeks of a core stability training program among badminton players. At the same time, Sharrock, Cropper, Mostad, Johnson and Malone [14] et al. found no correlation between core stability parameters and athletic performance parameters among tennis players.

The current study had some limitations. Firstly, we could not determine the existence of any correlation between height or weight and the other variables studied. Height and weight are important factors that may affect athletic-performance measures. Secondly, people with hamstring tightness were not excluded from the study. Tightness of the hamstring muscles can alter the pelvic position and can affect the activity of the abdominals in pelvic tilt control, which might have been masked in the study. Researchers may allow slight knee flexion in future studies during DLL performance to overcome hamstring bias in subjects with hamstring tightness. Upper-extremity muscle tightness could have affected the upper-extremity performance test scores in the current study, especially the UQYBT score. The flexibility of the participants’ upper-extremity muscles should have been verified before the tests. The motivational level of a subject may also affect his performance test scores. However, it was not checked in the current study, which could have had an impact on the test scores. Finally, the participants in the current study were healthy male collegiate athletes. Thus, the current study’s findings may not be directly transferable to female cohorts or professional athletes.

## 5. Conclusions

The results of the current study suggest that core stability measures, such as the McGill and DLL test scores, are positively correlated with most upper-extremity athletic-performance measures in collegiate athletes. The MBTT score was found to be the most significantly correlated with the scores in both core stability tests among the different upper-extremity performance test scores in the current study. However, due to the nature of the study, a cause−effect relationship cannot be established on the basis of the study’s findings, and the results should be interpreted with caution.

## Figures and Tables

**Table 1 medicina-58-00982-t001:** Demographic details of participants.

Variable	Mean ± SD.
Age (year)	23.6 ± 2.1
Height (cm)	174 ± 6.7
Body mass (kg)	72.3 ± 7.6
BMI (kg/m^2^)	24.3 ± 2.8
Weekly training hours	7.4 ± 0.71

BMI: Body mass index; data presented as mean ± SD.

**Table 2 medicina-58-00982-t002:** Descriptive statistics core stability and upper-extremity performance test scores performed by the participants. Values are expressed as mean ± SD.

Test	Value
Trunk extension test (S)	48.83 ± 6.92
Trunk flexion test (S)	110.6 ± 17.51
Right flexion test (S)	39.18 ± 11.20
Left flexion test (S)	57.86 ± 8.12
Total McGill score (S)	256.49 ± 38.60
Double leg lowering test (degree)	61.71 ± 5.22
Medicine ball throw test (cm)	178.20 ± 20.95
Upper quarter Y balance test (cm)	83.32 ± 14.60
Functional throwing performance index	42.89 ± 9.70

**Table 3 medicina-58-00982-t003:** Correlation between core stability test scores and upper-extremity performance test scores.

	McGill Score		DLL Score	
Upper-Extremity Performance Tests	*p*-Value	Pearson Correlation (r)	*p*-Value	Pearson Correlation (r)
Medicine ball throw test (MBTT)	0.02 *	0.67	0.02 *	0.25
Upper quarter Y balance test (UQYBT)	0.01 *	0.46	0.64	0.18
Functional throwing performance index (FTPI)	0.44	0.14	0.83	0.15

* Statically significant.

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
