# Peer review of "Correlation between Core Stability and Upper-Extremity Performance in Male Collegiate Athletes"

_medicina, 2022, doi:10.3390/medicina58080982_

Round 1
Reviewer 1 Report
Dear Authors,
Thank you for your effort in your work. After reading your research comprehensively and in detail, I, unfortunately, think that this research in its current form is not suitable for publication in the journal medicina. Below are my general and specific comments.
- First of all, the subject of your research is not original and has been studied a lot, and it also includes the generalization of athletes from different branches, not to a specific sports branch. (The branch of the athletes is very important here). Secondly, there are major deficiencies in all parts of the research, eg: The introduction is very short and simple, and you don't even have hypotheses in this part. I also found nothing to substantiate the originality of your research. In the materials and methods section you did not specify anything about the experimental design of your study (eg, were all the tests done on the same day? or in what order? did you do a randomized study? How reliable would it be to evaluate the results if all the tests were done on the same day? What were the rest intervals? i.e. . In your research, you need to strengthen the high correlations between core strength and upper extremity strength with regression analysis.
- In your discussion and conclusions section, you focused only on your own results and the results of other research. However, there is no information about the results that you attribute the similarities or differences between other studies and yours.
- Finally, I think that the number of your subjects is not enough to generalize for all athletes with the tests you used. If it was a specific branch, this number would be sufficient.
As a result, I recommend that you reshape your research with the suggestions I have given and, if necessary, increase the number of subjects in the study and resubmit it to a new journal with separate examinations in the branches.
I hope your research will be publishable after the necessary adjustments.
Yours sincerely
Author Response
I want to thank all the reviewers for providing us with an opportunity to rewrite the manuscript. Their comments and constructive suggestions helped us to improve this manuscript's quality. The reviewer's comments and authors' responses to the comments are given below.
- Reviewers comment :
First of all, the subject of your research is not original and has been studied a lot, and it also includes the generalization of athletes from different branches, not to a specific sports branch. (The branch of the athletes is very important here).
Authors' response to reviewer's comment:
Most of the studies to find out the correlation between core stability and sports performance investigated the lower limb performance parameters.Only a few studies available in the literature investigated the correlation between core stability and upper limb performance parameters.These studies do not provide accurate information on whether core stability parameters correlate with upper extremity performance parameters. Some of these studies reported a positive correlation and other studies didn’t show any correlation between these two parameters. According to Silfies, et al. [1], a true causal relationship between core training and upper-extremity performance cannot be inferred from any previous studies.
Regarding the generalisation of athletes author also agrees with the reviewer's comments. However, this will help in the gemeralisation of the study results. Future studies can be conducted for specific sports branch
- Reviewers comment
Secondly, there are major deficiencies in all parts of the research, e.g.: The introduction is very short and simple, and you don't even have hypotheses in this part. I also found nothing to substantiate the originality of your research.
Author's response to reviewer's comment:
The introduction has been updated by adding the hypothesis of the study
- Reviewers comment.
In the materials and methods section you did not specify anything about the experimental design of your study (e.g., were all the tests done on the same day? or in what order? did you do a randomized study? How reliable would it be to evaluate the results if all the tests were done on the same day? What were the rest intervals?
Authors' response to reviewer's comment:
The design of the study has been added to the methodology
The design of the study was a multivariate correlational design. The variables for core stability includes McGill and DLL test scores and the variables for upper-extremity performance include the test scores of UQYBT, MBTT and FTPI.
The tests were conducted at five stations in two different sessions with a 24-hour gap between the sessions. The tests for core stability were performed in the first session, and the tests for upper limb performance were taken in the second session. All the tests were performed in a random order in each session. All the participants completed both sessions in without experiencing any pain or injury.
A four-minute rest period was given between each test in each sessions . This has already been mentioned in the methodology.
- Reviewers comment
In your discussion and conclusions section, you focused only on your own results and the results of other research. However, there is no information about the results that you attribute the similarities or differences between other studies and yours.
Author's response to reviewer's comment:
The discussion section has been updated as per the suggestion
- Reviewers comment
Finally, I think that the number of your subjects is not enough to generalize for all athletes with the tests you used. If it was a specific branch, this number would be sufficient.
Author's response to reviewer's comment:
The sample size was calculated prior to the collection of the data which showed that 61 participants are required for the study. That is the reason study 61 participants were selected for the study .
Reviewer 2 Report
The Manuscript: „ Correlation between core stability and upper-extremity performance in collegiate athletes’’ by Shibili Nuhmani investigated the correlation of core stability with upper-extremity performance in collegiate athletes based on different standard tests. The authors identified a positive correlation between these two parameters. Previously, it has been shown that poor core stability increased the risk of upper extremity athletic injuries as well as it negatively affected athletic performance. Hence, the present study is just a follow-up of already established findings. The study is nicely conducted with elaborate description of methodology and documentation of subsequent result. After going through the manuscript, I have following comments to the author:
1. Core stability training has been frequently been suggested in injury prevention or performance enhancement programs for athletes. Please briefly discuss the significance of your present study on how it will be beneficial for athletes.
2. Only male participants were included in the study; hence, I would like to suggest the author to reflect this message in the title of the manuscript.
3. It has not been clearly mentioned in the manuscript how frequently were the test performed. Was it a one-time test on the athletes or multiple test were performed and the mean outcomes of individual athlete were used for the analysis.
Author Response
- Reviewers comment
Core stability training has been frequently been suggested in injury prevention or performance enhancement programs for athletes. Please briefly discuss the significance of your present study on how it will be beneficial for athletes.
Authors response to reviewer's comment :
The current study suggest that core stability measures are positively correlated with most upper-extremity athletic-performance measures in collegiate athletes. However as it is a correlational study a cause−effect relationship cannot be established on the basis of the study’s findings. An experimental study is warranted to establish a cause effect relationship .
- Reviewers comment
Only male participants were included in the study; hence, I would like to suggest the author to reflect this message in the title of the manuscript.
Authors' response to reviewer's comment :
Corrected as per the suggestion
- Reviewers comment
It has not been clearly mentioned in the manuscript how frequently were the test performed. Was it a one-time test on the athletes or multiple test were performed and the mean outcomes of individual athlete were used for the analysis.
Authors response to reviewer's comment :
The test was performed thrice and the average was taken for analysis .
Reviewer 3 Report
The purpose of this study was to investigate the correlation of core stability as measured by the McGill and double-leg lowering (DLL) test with upper-extremity performance as measured by the upper-quarter Y-balance test (UQYBT), medicine ball throw test (MBTT) and functional throwing performance index (FTPI) test in collegiate athletes.
From my point of view, the article is well written and easy to read. Although the thematic is not innovative, it is interesting and tries to provide answers to the existing questions raised in literature.
All in all, there is nothing to say or add in relation to the various sections of the study as it is written clearly, concisely and well organized. However, the authors should pay attention to minor points, such as in the results where the “p” and “r” values do not follow a consistent decimal place (e.g. P = 0.023, r = 0.671 / P = 0.02, r = 0.25).
Moreover, the final references list should be revised considering that some references do not follow the guidelines of the journal.
Author Response
- Reviewers comment
From my point of view, the article is well written and easy to read. Although the thematic is not innovative, it is interesting and tries to provide answers to the existing questions raised in literature.
Authors' response to reviewer's comment
Thank you for your comments
- Reviewers comment
All in all, there is nothing to say or add in relation to the various sections of the study as it is written clearly, concisely and well organized. However, the authors should pay attention to minor points, such as in the results where the “p” and “r” values do not follow a consistent decimal place (e.g. P = 0.023, r = 0.671 / P = 0.02, r = 0.25).
Authors response to reviewer's comment
Corrected as per the suggestion
- Reviewers comment
Moreover, the final references list should be revised considering that some references do not follow the guidelines of the journal.
Authors' response to reviewer's comment
Corrected as per the suggestion
Round 2
Reviewer 1 Report
Dear Author,
Thank you for your revised manuscript. I think your manuscript need some revision, but in current form it is acceptable for journal.
Congrulations.
Best